# Soil Chemical Quality in Integrated Production Systems with the Presence of Native and Exotic Tree Components in the Brazilian Eastern Amazon

Ivanderlete Marques de Souza [1], Edvaldo Sagrilo [2,*], José Oscar Lustosa de Oliveira Júnior [2], Maria Diana Melo Araújo [1], Luciano Cavalcante Muniz [3], Joaquim Bezerra Costa [4], Roberto Cláudio Fernandes Franco Pompeu [5], Daiane Conceição de Sousa [6], Hosana Aguiar Freitas de Andrade [7], Edson Dias de Oliveira Neto [7], Luiz Fernando Carvalho Leite [2], Flávio Favaro Blanco [2], Paulo Sarmanho da Costa Lima [2] and Henrique Antunes de Souza [2]

1   Center of Agrarian and Bilogical Sciences, Universidade Estadual Vale do Acaraú, Campus Sobral, 850 Da Universidade Ave., Sobral 62010-295, CE, Brazil; ivanderlete@gmail.com (I.M.d.S.); mariadianamello@gmail.com (M.D.M.A.)

2   Embrapa Meio-Norte, Duque de Caxias Av. 5650, Bairro Buenos Aires, Teresina 64008-780, PI, Brazil; jose.oscar@embrapa.br (J.O.L.d.O.J.); luiz.f.leite@embrapa.br (L.F.C.L.); flavio.blanco@embrapa.br (F.F.B.); paulo.costa-lima@embrapa.br (P.S.d.C.L.); henrique.souza@embrapa.br (H.A.d.S.)

3   Agronomy Studies Unit, Department of Rural Economy, Universidade Estadual do Maranhão, 1000 Lourenço Vieira da Silva Ave., São Luis 65055-970, MA, Brazil; luciano-muniz@uol.com.br

4   Embrapa Cocais, São Luís Rei de França Av., São Luís 65020-500, MA, Brazil; joaquim.costa@embrapa.br

5   Embrapa Caprinos e Ovinos, Sobral-Groaíras Road Km 4, Sobral 62010-970, CE, Brazil; roberto.pompeu@embrapa.br

6   Center of Agroforestry Sciences, Universidade Federal do Sul da Bahia, Itabuna 45600-923, BA, Brazil; dcsousa.solum@gmail.com

7   Department of Agronomy, Universidade Federal do Piauí, Campus Ministro Petrônio Portela, W/N–Ininga, Teresina 64049-550, PI, Brazil; hosanaguiarf.andrade@gmail.com (H.A.F.d.A.); edson_neto@live.com (E.D.d.O.N.)

*   Correspondence: edvaldo.sagrilo@embrapa.br; Tel.: +55-86999071896

**Abstract:** Conservation systems involving trees enhance the sustainability of tropical soils. However, little is known on the effect of integrated systems with native and exotic trees on soil chemical quality in the eastern Amazon. We aimed to measure changes in soil chemical quality in integrated production systems in Pindaré-Mirim, Maranhão, Brazil. This study was carried out in 2017 and 2018, evaluating (i) perennial pasture; (ii) crop–livestock–forest integration-I (CLFI-I)—eucalyptus rows interspersed with maize + *Urochloa brizantha* intercropping; (iii) CLFI-II—babassu palm trees (*Attalea speciosa* Mart.) with maize + *Megathyrsus maximus* intercropping; and (iv) maize + *M. maximus* intercropping. Soil chemical attributes at depths of 0.00–0.10 m, 0.10–0.20 m, 0.20–0.30 m, and 0.30–0.50 m, forage productivity, and soil cover were evaluated. CLFI-II promoted the highest soil organic matter concentration in topsoil and highest pH, lowest $Al^{3+}$ levels, and potential acidity (H+Al) at all soil depths. Soil under pasture showed the highest N, $K^+$, $Ca^{2+}$ concentrations, sum of bases, and cation exchange capacity. Changes in CLFI-II are associated with the babassu palm's ability to modulate the surrounding environment, giving the species a competitive advantage in anthropic environments. The time of adoption is crucial for improving soil fertility in the Brazilian eastern Amazon. Sustainable production systems in the region must comply with long-term management plans.

**Keywords:** *Attalea speciosa* Mart.; forage grasses; intercropped maize; soil cover; soil fertility; soil organic matter

## 1. Introduction

The Brazilian Amazon is the largest tropical rainforest in the world [1] and can be divided into two large portions: the western Amazon and the eastern Amazon [2]. The east-

ern Amazon includes Brazilian states that form a zone known as the "Arc of Deforestation", often associated with the deforestation of native forests [3,4]. In Maranhão, a Brazilian state included in this arc, the accumulated deforestation rate in 2023 was 189.19 km$^2$ [5]. The clearing of native forests drastically affects biodiversity [6] and reduces the functionality of ecosystem services promoted by tropical forests [7,8].

The main cause of rainforest deforestation is linked to the demand for agricultural commodities. Areas previously occupied by a natural ecosystem and protected from human modifications have been converted into soybean fields and pasture for livestock [9]. Areas naturally covered by native and dense vegetation allow the efficient sequestration and cycling of nutrients in the Amazonian soils through litter production [10]. However, extensive pasture management can result in environmental degradation, such as losses in land cover. Tropical soils are highly weathered and poor in nutrients; thus, pasture management without adequate fertilization to replenish extracted nutrients quickly leads to soil exhaustion [11].

Inadequate management is one of the main causes of the reduced agricultural productivity and degradation of agroecosystems in the Amazon. The exhaustion of productive capacity in degraded areas can lead to the abandonment of land [12] for the search for new areas for deforestation and the restart of a new productive cycle. Conversely, the pressure caused by deforestation and the conversion of native forests for agricultural activities can be reduced by recovering soil in older areas, especially degraded pastures [13].

Efforts have recently been made to develop sustainable approaches in the Amazon for the intensification of livestock farming in deforested areas [14]. To mitigate the impacts caused by traditional production systems, it is important to explore spatial land-use arrangements to improve the provision of ecosystem services. Within this approach, integrated production systems have often been associated with multiple benefits for soil health [15]. Integrated systems include the intercropping of grain crops such as maize with tropical forage grasses. This intercrop is capable of producing a high quantity of roots and aboveground biomass, which significantly improves soil quality and allows the inclusion of the animal component in agricultural areas, enabling sustainable intensification of production. Studies have revealed evidence of increased nutrient availability in integrated systems [16] mediated by litter deposition, leading to an increase in soil organic matter (SOM). SOM represents a potential, dynamic source of nutrients for soil [17]. In these systems, species diversification makes nutrient cycling and use more efficient [18].

Forest species such as eucalyptus are frequently included in integrated production system arrangements in Brazil to meet the demand for fibers and energy [19]. Furthermore, these species also stand out as a key component in soil conservation practices [20,21]. Due to eucalyptus' rapid growth, rusticity, and adaptability to Brazilian soils, the tree is the main species used in integrated production systems in the country [22]. However, there is a growing need to investigate the effect of including native species from the primary forest in the integrated systems, such as the babassu palm (Attalea speciosa Mart.). This species has unparalleled social and economic importance for smallholders in the eastern Amazon due to the exploitation of many products from extractivism [23]. Moreover, this species has great ecological importance because of its resilience and abundance even after vegetation fires and soil degradation in the Amazon [24]. Babassu palm is a ruderal species, with wide occurrence and with a dominance ability in anthropized areas located in the "Arc of Deforestation" of the Amazon.

Little is known about the impacts of tree components on crop–livestock–forest integration systems (CLFIs) in the Amazon region, the use of babassu palm as a tree component to replace eucalyptus in integrated production systems, and their effects on soil fertility in degraded areas of the region. Therefore, we tested the following hypotheses: (i) the presence of the tree component improves the chemical attributes of soils in the Amazon region in comparison with systems without the tree component; and (ii) the babassu palm can be used as a native tree component to replace eucalyptus in integrated production systems. To test these hypotheses, this study aimed to measure changes in the chemical

quality of soil and the production of plant biomass in integrated production systems in the eastern portion of the Brazilian Amazon, in the state of Maranhão.

## 2. Materials and Methods

### 2.1. Climate and Soil of the Experimental Area

This study was carried out from January 2017 to February 2018 in a Technological Reference Unit (TRU) located in the Pindaré-Mirim microregion, state of Maranhão, Brazil (03°46′ S, 45°29′ W, and 38 m above sea level) (Figure 1).

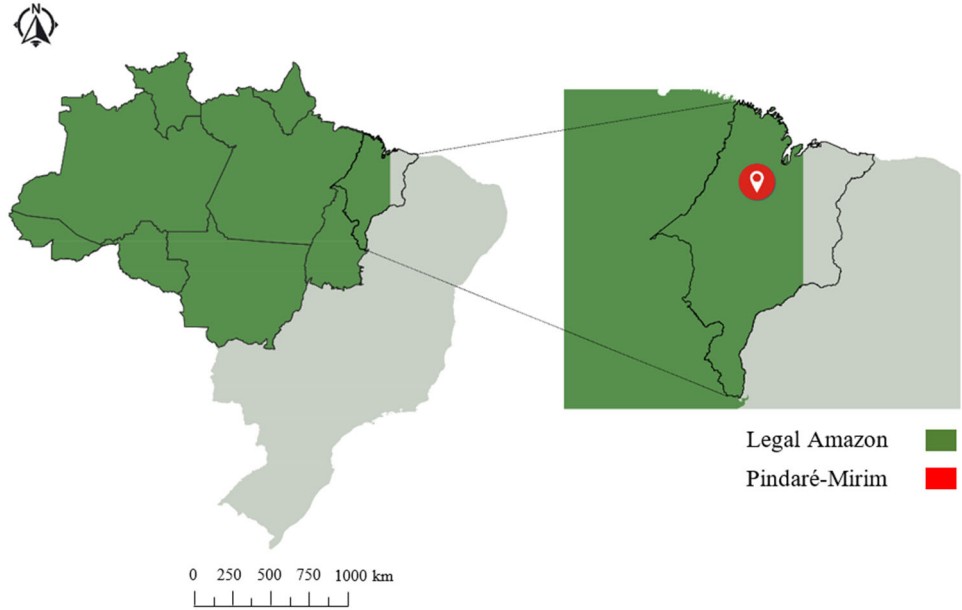

**Figure 1.** Geographic location of the municipality of Pindaré-Mirim, Maranhão, Brazil.

The region's climate is classified as Aw (Tropical, hot, and humid), according to the Köppen classification [25]. The average annual precipitation during the experimental period was 2570 mm, concentrated between January 2017 and February 2018 (Figure 2). The soil in the study region is classified as Haplic Plinthosol, according to the Brazilian Soil Classification System [26], and Plinthosol according to the World Reference Base [27].

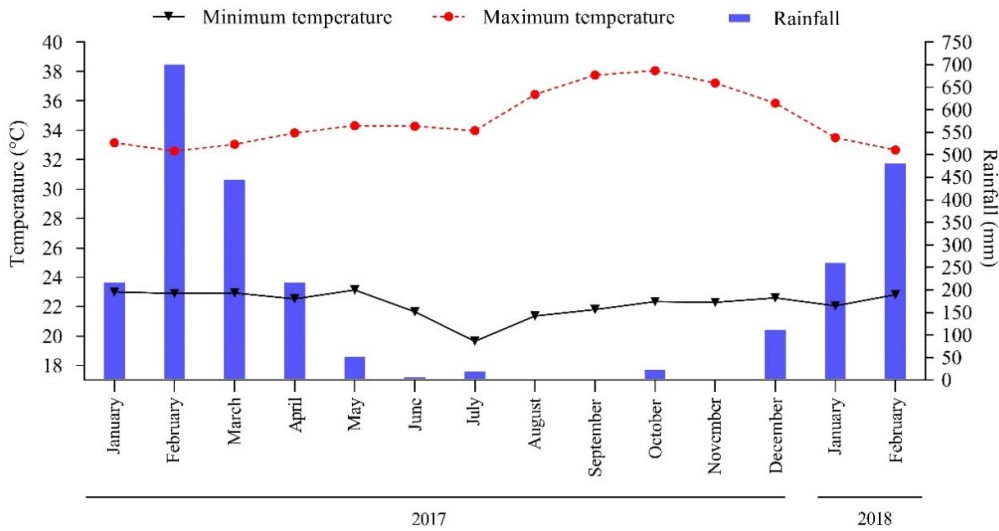

**Figure 2.** Monthly total rainfall and maximum and minimum temperature in Pindaré-Mirim, Maranhão, Brazil.

## 2.2. History of the Areas and Description of the Systems

Four land use systems were evaluated in an area with a history of degradation, corresponding to the following: (1) perennial pasture with 17 years of *Urochloa brizantha* cv. Marandú—reference system; (2) integrated crop–livestock–forest I (CLFI-I)—composed of eucalyptus rows (forest component) and inter-rows with maize intercropped with *Urochloa brizantha* cv. Marandú (crop component); (3) integrated crop–livestock–forest system II (CLFI-II)—composed of babassu palm trees (forest component) in a Savanna-type arrangement associated with maize intercropped with *Megathyrsus maximum* cv. Massai; and (4) maize intercropped with *Megathyrsus maximum* cv. Massai. The areas of each system had dimensions of 3.0 ha (pasture, ICLF-II, and maize + Massai intercropping) and 3.5 ha (ICLF-I), distributed adjacently in the field. A detailed description and history of the management systems are presented in Table 1.

**Table 1.** Description and history of agricultural systems in Pindaré-Mirim, Maranhão state, Brazil.

| Land Use | Historic |
|---|---|
| Perennial Pasture | Previously (1970–1999), Jaraguá grass (*Hyparrhenia rufa* (Ness) Stapf) was cultivated. After pasture renewal (2000–2017), it consisted of an area of 3.0 ha cultivated with *Urochloa brizantha* cv. Marandú, with no soil correction or fertilization, used for continuous extensive grazing with a stocking rate of 0.5 animals ha$^{-1}$. The absence of management practices allowed the natural regeneration of native vegetation. |
| ICLF-I | Area of 3.5 ha with 2 years of maize cultivation (hybrids KWS 9304 and AG1051) in a spacing of 0.6 m $\times$ 0.3 m (66,600 plants ha$^{-1}$), intercropped with *Urochloa brizantha* cv. Marandú. The tree component (eucalyptus) was implemented in February 2017, in double rows with a spacing of 3 m $\times$ 2 m and 28 m between rows. |
| ICLF-II | Area of 3.0 ha, with native babassu palm vegetation (average of 35 palm trees ha$^{-1}$). Maize (Dow Herculex) intercropped with Massai grass (*Megathyrsus maximum*) was implemented in February 2017. Maize was sown at a spacing of 0.6 m $\times$ 0.3 m (55,000 plants ha$^{-1}$). A cultural value of 98% was considered for sowing the Massai seeds. Before the implementation of the intercropping, the area was cultivated with *Urochloa brizantha* cv. Marandú for 20 years. |
| Maize + Massai intercropping | Area of 3.0 ha, with maize (Dow Herculex) planted in January 2017 intercropped with Massai grass (*Megathyrsus maximum*). Maize was sown at a spacing of 0.5 m $\times$ 0.25 m (83,000 plants ha$^{-1}$). A cultural value of 98% was considered for sowing the Massai seeds. Before the implementation of the intercropping, the area was cultivated with *Urochloa brizantha* cv. Marandú for 20 years. |

Before sowing, 1.8 Mg ha$^{-1}$ of dolomitic limestone (85% of relative neutralization value—RNV; >12% MgO) was applied to soil in all systems, except for the perennial pasture. Dolomitic limestone was used due to the low Mg$^{2+}$ concentrations in the soil of the experimental area. Base fertilization of maize was carried out using 400 kg ha$^{-1}$ of 05-30-15 (NPK). A total of 200 kg ha$^{-1}$ of 36-00-30 was also applied as top dressing 10 days after maize emergence and 200 kg ha$^{-1}$ of 36-00-30 20 days after the first top dressing. Liming and fertilizers were applied according to soil analysis and following recommendations for the region [28]. For the sowing of grasses in the pasture treatments, CLFI-I, and CLFI-II, 10 kg ha$^{-1}$ of seeds were used in a specific box compartment in the planter. In the treatment of maize + forage grass intercropping, Massai grass seeds (10 kg ha$^{-1}$) were mixed with the fertilizer used in the base fertilization (400 kg ha$^{-1}$ of 05-30-15 NPK formulation). As for eucalyptus, at the time of planting, 0.075 kg of phosphate was applied to each pit at a depth of 0.30 m and 0.15 kg of the 36-00-30 NPK formulation.

### 2.3. Soil Sampling and Analysis

Soil samples were collected in each system at the end of the rainy season, in June 2017. In each area, four sampling points were established randomly every 50 m. At each of these points, twelve single soil samples were collected at depths of 0–0.10 m, 0.10–0.20 m, 0.20–0.30 m, and 0.30–0.50 m, distributed in all cardinal directions. From the 12 single samples collected at each depth, a composite sample (replicate) containing approximately 300 g was formed. Therefore, four composite samples (replicates) were obtained in each management system and at each depth. In areas with a tree component (babassu palm and eucalyptus), half of the single samples at each point were collected under the tree canopy (approximately 1.5 m away), and the rest of the single samples were collected in an area without influence of the tree canopy.

The collected soil was used to determine pH in water (soil/solution ratio 1:2.5). SOM was quantified after digestion using $K_2Cr_2O_7$ mol $L^{-1}$ and titration with $Fe_2SO_4$ 0.2 mol $L^{-1}$. Exchangeable $Ca^{2+}$ and $Mg^{2+}$ were extracted with 1 mol $L^{-1}$ KCl and determined by complexometric titration. Exchangeable $Na^+$ and $K^+$ were extracted using 0.05 mol $L^{-1}$ HCl solution and determined by flame photometry. Exchangeable $Al^{3+}$ was extracted with 1 mol $L^{-1}$ KCl solution and determined by titration. Potential acidity (H+Al) was determined after extraction with a 0.5 mol $L^{-1}$ calcium acetate solution buffered at pH 7.0 and quantified by titration. Available P was extracted with Mehlich-1 solution and determined by colorimetry. Based on the results of the chemical analysis, the values of the sum of bases (SB), cation exchange capacity (CEC), and base saturation (BS) were calculated. All chemical analyses were performed according to procedures described by Teixeira et al. [29].

Samples of the aerial part of forage were collected in February 2018 in the different systems to estimate forage productivity. This period was chosen for sampling because it coincides with the rainy season in the region, when the aerial part of plants is fully developed. A metal frame of known area (0.25 $m^2$) was used to delimit the perimeter of the plant biomass harvested at the cutting height recommended by Pires [30] for each species. After collection, the material was weighed and dried in a forced air circulation oven (65 °C) until constant mass. At the same time as field sample collection, the assessment of vegetation cover was also carried out through visual estimation, according to the procedure described by Gazziero et al. [31]. The method considers the average score from two different evaluators, establishing a rating scale ranging from 0 to 100%, where "zero" refers to the absence of vegetation cover, and "100" refers to total soil coverage. The area for evaluating vegetation cover in each system was delimited by a metal frame (0.25 $m^2$) before cutting the forage.

### 2.4. Statistical Analyses

Data relating to vegetation cover, forage production, and soil chemical attributes were used to create boxplots, and any outliers were identified and eliminated. The data from the replicates of the variables for each management system were used to determine the mean and establish the confidence interval (CI) of the mean ($p < 0.05$), according to Payton et al. [32]. In this method, when the upper and lower limits of the confidence interval do not overlap, it is considered that there is a significant difference between the different treatments. Data on soil chemical attributes at each depth were also subjected to principal component and cluster analyses, after standardizing the variables [33]. All analyzes were carried out using the version 3.6.0+ of the R statistical package, [34].

## 3. Results

There was a general trend of highest pH values and lowest potential acidity values (H+Al) and $Al^{3+}$ concentration in soil under CLFI-II. The CLFI-II management system and pasture were also responsible for the highest SOM levels at a depth of 0.00–0.10 m (Figure 3).

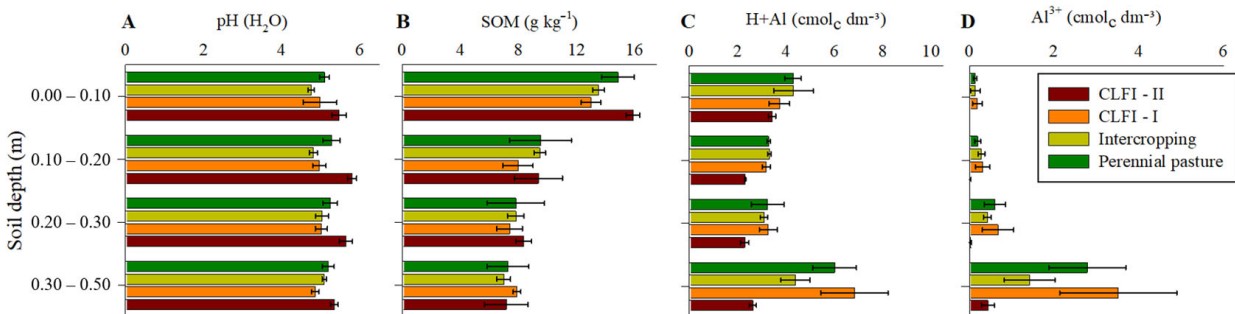

**Figure 3.** Averages and confidence intervals ($p < 0.05$) of the pH values (**A**), SOM concentration (**B**), potential acidity (H+Al) (**C**), and $Al^{3+}$ (**D**) for perennial pasture, maize + Massai intercrop, CLFI-I, and CLFI-II, in the soil depths of 0.00–0.10 m, 0.10–0.20 m, 0.20–0.30 m, and 0.30–0.50 m. Pindaré-Mirim, Maranhão, Brazil.

Considering the lower and upper limits of the confidence intervals, a higher N concentration in soil was observed in the area under pasture in the 0.00–0.10 m layer (Figure 4). In turn, the soil P concentration found in the pasture area was lower than that observed in other management systems, at all depths. Overall, highest $K^+$ levels were observed in the areas under pasture and CLFI–I at all depths. $Na^+$ concentrations were highest in the CLFI–II system in the 0.00–0.10 layer and in the CLFI-II and pasture systems in the 0.10–0.20 m layer. For $Ca^{2+}$ concentrations, there was a tendency for highest values in the area under pasture at all soil depths. In turn, $Mg^{2+}$ concentrations were significantly lower in the area under CLFI-I at all depths, compared to other management systems (Figure 4).

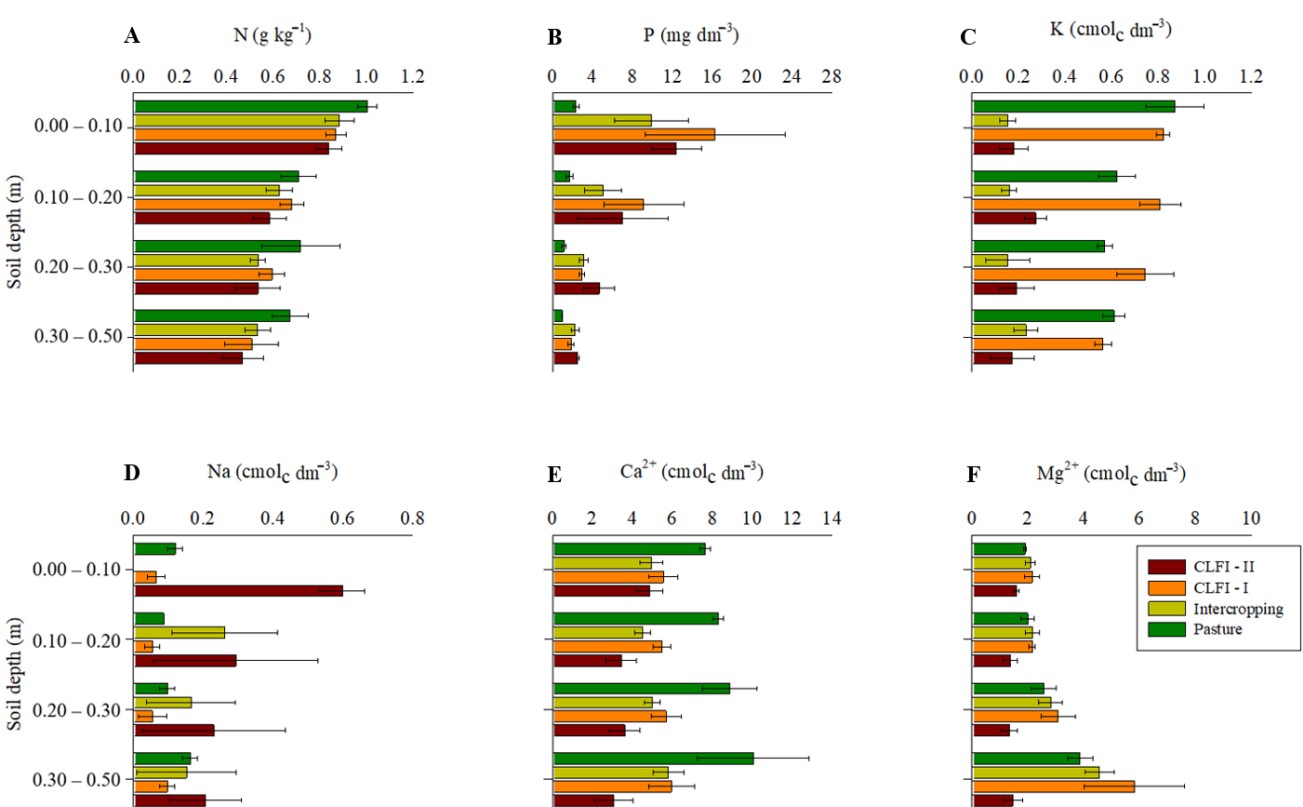

**Figure 4.** Averages and confidence intervals ($p < 0.05$) for the soil concentrations of N (**A**), P (**B**), $K^+$ (**C**), $Na^+$ (**D**), $Ca^{2+}$ (**E**), and $Mg^{2+}$ (**F**) in perennial pasture, maize + Massai intercrop, CLFI-I, and CLFI-II, in the soil depths of 0.00–0.10 m, 0.10–0.20 m, 0.20–0.30 m, and 0.30–0.50 m. Pindaré-Mirim, Maranhão, Brazil.

Highest values of SB and CEC were observed in the pasture, at depths of 0.00–0.10 and 0.10–0.20 m. The system under pasture did not differ from CLFI-I at a depth of 0.20–0.30 m and from CLFI-I and the maize + Massai intercropping at a depth of 0.30–0.50 m. Regarding BS values, no differences were observed between the systems (Figure 5).

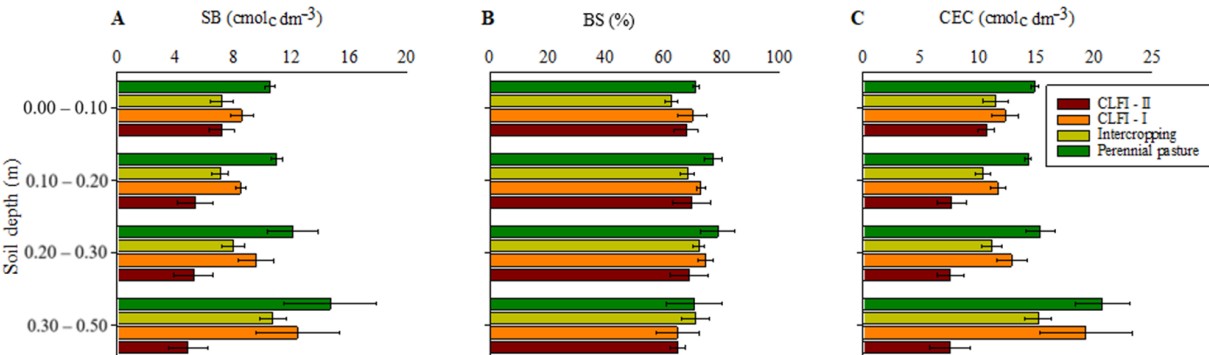

**Figure 5.** Averages and confidence intervals ($p < 0.05$) for the SB (**A**), BS (**B**), and CEC (**C**) for soil under perennial pasture, maize + Massai intercrop, CLFI-I, and CLFI-II, in the soil depths of 0.00–0.10 m, 0.10–0.20 m, 0.20–0.30 m, and 0.30–0.50 m. Pindaré-Mirim, Maranhão, Brazil.

Soil coverage in the CLFI-I system (98.2%) was higher than those of other management systems, followed in descending order by pasture (58.1%), maize + Massai intercropping (36.6%), and CLFI-II (34.4%). Forage dry mass productivity values were not different between the systems, with an average value of 1480 kg ha$^{-1}$ (Figure 6).

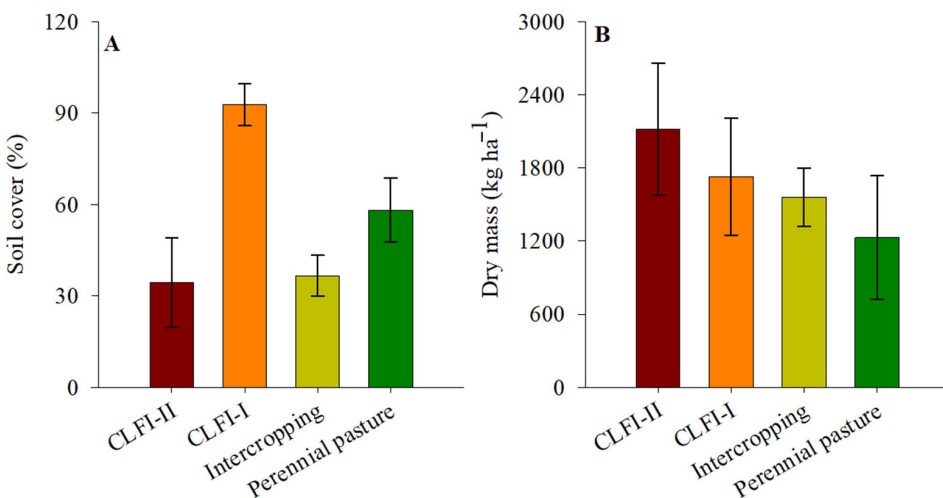

**Figure 6.** Averages and confidence intervals ($p < 0.05$) for the variables soil cover (**A**) and dry mass forage production (**B**) under perennial pasture, maize + Massai intercrop, CLFI-I, and CLFI-II, in the soil depths of 0.00–0.10 m, 0.10–0.20 m, 0.20–0.30 m, and 0.30–0.50 m. Pindaré-Mirim, Maranhão, Brazil.

PCA demonstrated that the two components (PC1 and PC2) together explained 85.92, 83.20, 95.34, and 90.86% of the total dissimilarity between managements at depths of 0.00–0.10 (Figure 7A), 0.10–0.20 (Figure 7B), 0.20–0.30 (Figure 7C), and 0.30–0.50 m (Figure 7D), respectively. The correlation between the variables and the principal components was considered relevant when the coefficient weights were greater than 0.3 (Table 2). The CLFI-II system correlated with the attributes pH, Na$^+$, and SOM at depths of 0.00–0.10, 0.10–0.20, and 0.20–0.30 m and with P at a depth of 0.30–0.50 m. In general, the area under pasture correlated with N, K$^+$, Ca$^{2+}$, SB, and CEC, while the CLFI-I system and the maize + Massai intercropping were associated with Mg$^{2+}$, H+Al, and Al$^{3+}$.

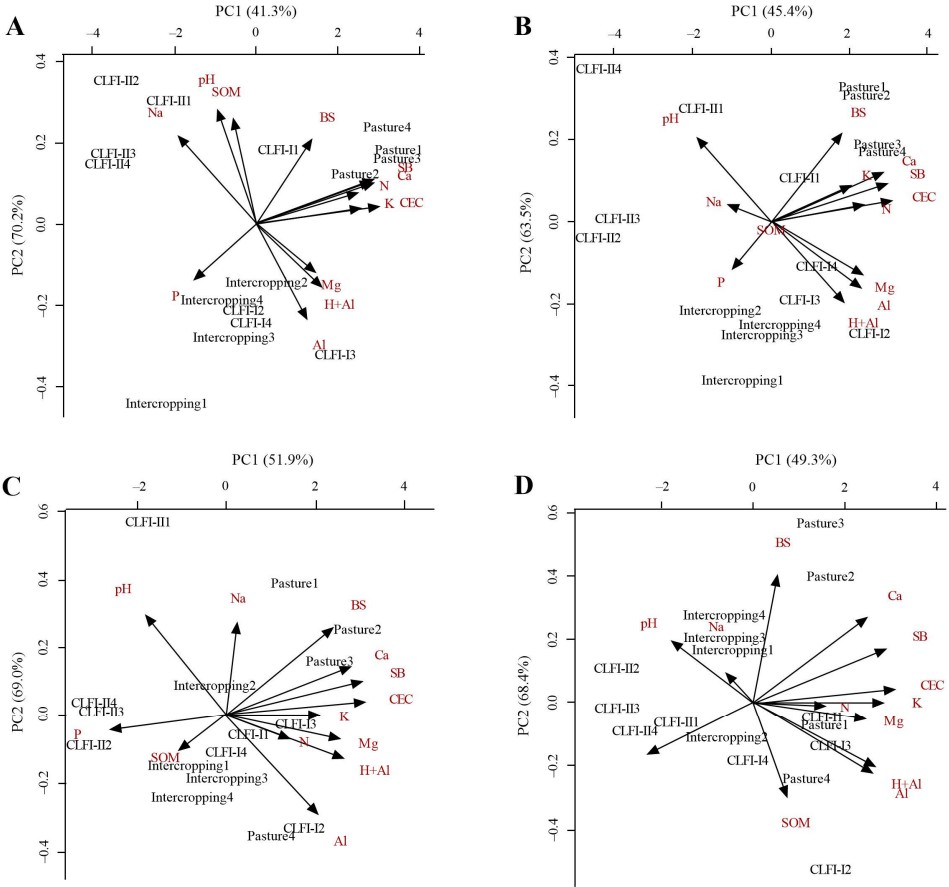

**Figure 7.** Biplot of the relationship between soil attributes and different soil management systems (perennial pasture, maize + Massai intercrop, CLFI-I, and CLFI-II) for the two principal components (PC1 and PC2), in the soil depths of 0.0–0.10 m (**A**), 0.10–0.20 m (**B**), 0.20–0.30 m (**C**), and 0.30–0.50 m (**D**). Pindaré-Mirim, Maranhão, Brazil.

**Table 2.** Weight coefficients (eigenvectors), eigenvalues, and variances of the soil chemical attributes and different management systems, explained by each principal component (PC1 and PC2) in the soil depths of 0.00–0.10 m, 0.10–0.20 m, 0.20–0.30 m, and 0.30–0.40 m. Pindaré-Mirim, Maranhão, Brazil.

| Chemical Attributes | Soil Depths (m) | | | | | | | |
|---|---|---|---|---|---|---|---|---|
| | 0.00–0.10 | | 0.10–0.20 | | 0.20–0.30 | | 0.30–0.50 | |
| | PC1 | PC2 | PC1 | PC2 | PC1 | PC2 | PC1 | PC2 |
| pH | −0.13 | 0.45 | −0.25 | 0.44 | −0.22 | 0.48 | −0.22 | 0.27 |
| SOM | −0.08 | 0.42 | 0.00 | −0.03 | −0.13 | −0.17 | 0.09 | −0.41 |
| N | 0.34 | 0.13 | 0.31 | 0.07 | 0.17 | −0.10 | 0.20 | −0.01 |
| P | −0.21 | −0.22 | −0.13 | −0.25 | −0.32 | −0.07 | −0.29 | −0.23 |
| $K^+$ | 0.35 | 0.06 | 0.26 | 0.18 | 0.26 | 0.00 | 0.35 | 0.00 |
| $Na^+$ | −0.26 | 0.35 | −0.15 | 0.08 | 0.03 | 0.45 | −0.08 | 0.14 |
| $Ca^{2+}$ | 0.39 | 0.17 | 0.36 | 0.25 | 0.34 | 0.23 | 0.31 | 0.37 |
| $Mg^{2+}$ | 0.20 | −0.19 | 0.30 | −0.28 | 0.31 | −0.11 | 0.31 | −0.07 |
| H+Al | 0.22 | −0.25 | 0.24 | −0.43 | 0.32 | −0.20 | 0.33 | −0.28 |
| $Al^{3+}$ | 0.17 | −0.38 | 0.30 | −0.35 | 0.25 | −0.47 | 0.32 | −0.31 |
| SB | 0.39 | 0.18 | 0.39 | 0.20 | 0.37 | 0.16 | 0.36 | 0.23 |
| CEC | 0.41 | 0.07 | 0.40 | 0.10 | 0.38 | 0.07 | 0.39 | 0.06 |
| BS | 0.19 | 0.34 | 0.23 | 0.45 | 0.29 | 0.42 | 0.07 | 0.56 |
| Eigenvalues | 5.36 | 3.76 | 5.90 | 2.35 | 6.75 | 2.21 | 6.41 | 2.47 |
| Total variance (%) | 41.3 | 28.9 | 45.4 | 18.1 | 51.9 | 17.1 | 49.3 | 19.1 |
| Cumulative variance (%) | 41.3 | 70.2 | 45.4 | 63.5 | 51.9 | 69.0 | 49.3 | 68.4 |

The management systems were grouped based on the degree of similarity, with CLFI-II forming a distinct group from the other management systems (CLFI-I, maize + Massai intercropping, and pasture) at all soil depths (Figure 8).

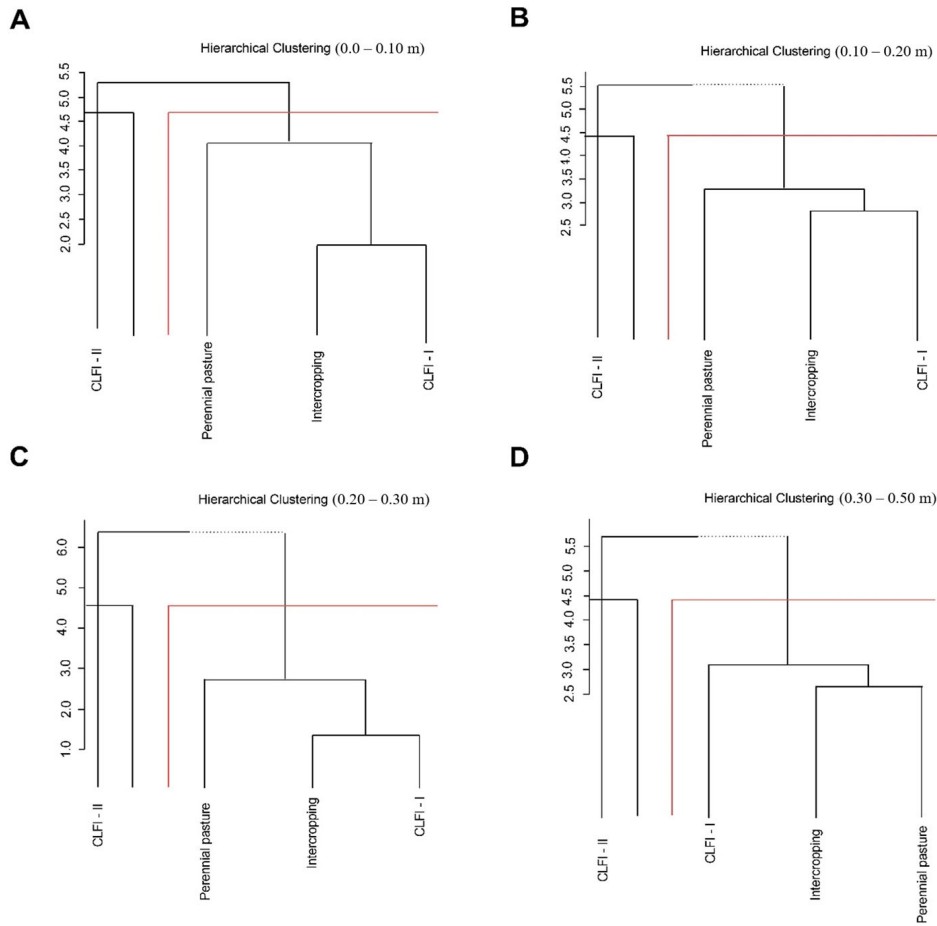

**Figure 8.** Dissimilarity dendrogram between soil attributes and different soil management systems (perennial pasture, maize + Massai intercrop, CLFI-I, and CLFI-II), in the soil depths of 0.0–0.10 m (**A**), 0.10–0.20 (**B**), 0.20–0.30 (**C**), and 0.30–0.50 m (**D**).

## 4. Discussion

Alongside the area under perennial pasture, the CLFI-II system led to the highest SOM concentration in topsoil (0–10 cm). These results are consistent with studies demonstrating the ability of babassu palm to increase soil organic carbon (SOC) stocks in integrated systems in the eastern portion of the Amazon [35]. Babassu palm has wide occurrence in anthropized areas of the Amazon [36]. The high carbon to nitrogen ratio (C:N) and slow decomposition rate of its biomass [24] is generally associated with high levels of lignin [37], whose incomplete decomposition is a precursor in the formation of SOM [38]. The integrated system containing babassu palm (CLFI-II) demonstrated great ability to produce biomass, although it was not uniformly distributed over the soil surface. Babassu palm has a strategic competitive advantage because of its litter production with a high C:N ratio that accumulates around the plants, resulting in an increase in surface SOC stocks [39]. This suggests that a considerable part of the biomass produced in the CLFI-II system originated from babassu plants and remained concentrated under the trees' canopies. Thus, the absence of soil disturbance close to the plants, associated with the continuous supply of litter present at different stages of decomposition, led to an increase in SOM levels. Other authors [40] have demonstrated that the permanent addition of litter to soil increases carbon

concentration, mainly due to increases in the light fraction of organic matter composed of material in the initial stages of decomposition [41,42].

Similarly, roots play an important role in SOM accumulation. It was demonstrated [43] that the roots of primary vegetation species, with a predominance of babassu palm, are concentrated close to the soil surface (46.0% of roots in the 0.00–0.10 m layer and 88.0% in the layer up to 0.40 m), which may have contributed to a greater incorporation of SOM in the most superficial layer. Furthermore, in the plant's zone of influence, babassu palm has a disproportionate dominance of roots compared to other species. Roots typically have more lignified cells than the aerial part of plants [44], which reduces their decomposition rate and contributes to a greater incorporation of SOM [45].

Up to 0.30 m deep, $Al^{3+}$ ions corresponded to a small fraction of the potential acidity (H+Al), in contrast to their larger contribution in the deepest layer evaluated (0.30–0.50 m), as there was a reduction in SOM concentrations in all treatments. It is possible that the higher SOM concentrations in more superficial layers contributed in a greater proportion to the potential acidity through the supply of $H^+$ ions from the organic components of SOM [46]. However, the contribution of SOM does not explain the highest soil pH values in the treatment with the presence of babassu palm (CLFI-II), especially in the 0.00–0.10 m layer, where this treatment also led to the highest SOM values. Mineral N fertilization can lead to soil acidification. However, no annual crops were cultivated under the babassu's plant canopies, and therefore, no mineral N was applied in sampling points close to babassu plants distributed across the area of ICLF-II. This may have contributed to higher pH in this system compared to the others. Moreover, studies suggest that the babassu palm tree can modify the surrounding soil environment through the deposition of litter by changing SOM and nutrient concentrations, thereby creating conditions that favor the tree's dominance [39]. Thus, it is possible that the modulation of pH by babassu palm gives this species the ability to establish itself as having competitive advantages in an anthropically degraded environment [47]. Despite this, evidence about the mechanisms associated with the increase in soil pH in the system with the presence of babassu palm remains unknown, mainly because lime application in the CLFI-II system was similar to that of the CLFI-I and maize + Massai intercropping systems.

Despite the highest pH values in the area under CLFI-II, this system showed lower concentrations of $Ca^{2+}$ and $Mg^{2+}$, in addition to lower SB and CEC, especially compared to the pasture area. Teixeira et al. [48] state that cropping systems with high $Ca^{2+}$ concentrations and base saturation normally do not correlate positively with high SOM concentrations, which is partially in line with the results of this study. However, the authors suggest a direct correlation between higher $Ca^{2+}$ concentrations and base saturation with higher pH values, modulated by the application of lime, which was not confirmed by the results of the present study. Although the area under perennial pasture was not limed, it is possible that the long period of adoption of this system contributed to the high $Ca^{2+}$ and BS values because forage grasses could increase these two soil attributes [49].

Continuous pasture cultivation over 17 years also resulted in a higher N concentration in topsoil (0.00–0.10 m), which may be associated with the ability of pastures to accumulate N in their biomass, especially in leaves [50]. Additionally, there is the permanent addition of N to soil, originating from animal waste. Approximately 90% of N contained in pastures returns to soil in the form of animal urine or feces [51]. Of this total, between 40 and 50% is reused by the pasture, remaining in the system, with the remainder lost mainly through the volatilization of $NH_3$ [52]. Conversely, although the CLFI-I, CLFI-II, and maize + Massai intercropping systems received fertilization with N sources, the main product of these systems (maize grains) was harvested, resulting in lower soil N concentrations. Studies [53] have demonstrated that 74% of all N absorbed by maize plants accumulate in the grains, which is equivalent to approximately 138.2 kg ha$^{-1}$. Similar to N, much of the P absorbed by maize plants is reallocated to the grains (84%) and removed from the field through harvest, which is equivalent to approximately 4.53 kg ton$^{-1}$ of grains produced [53]. However, unlike N, a considerable part of the P applied in the form of fertilizer is not absorbed by

plants, accumulating in the soil [54], which explains the highest P concentrations in CLFI-I, CLFI-II, and maize + Massai intercropping. This fact arises from the high levels of iron oxides ($Fe^{2+}$) and $Al^{3+}$ typically observed in tropical soils, resulting in the P adsorption and causing only a fraction of the P to be effectively available for uptake by plants [55].

The areas under perennial pasture and the integrated system with eucalyptus (CLFI-I) promoted the highest soil $K^+$ concentrations. Grasses of the genus *Urochloa* have a massive root system [20]. Thus, the earlier established (17 years) area under Marandu grass may have contributed to better $K^+$ cycling, resulting in high soil $K^+$ concentrations. Similarly, eucalyptus played a relevant role in $K^+$ cycling in soil, possibly influenced by the high $K^+$ concentrations in the trees' leaves, bark, and branches [56]. This effect was also expected for the system containing babassu palm (CLFI-II), mainly because this species has leaves with high $K^+$ concentrations [24], influencing soil $K^+$ concentrations, especially under the trees' canopies. However, contrary to expectations, this result was not observed in the present study. Unlike $K^+$, high $Na^+$ concentrations were observed in soil in the CLFI-II system, especially at a depth of 0.00–0.10 m. These data contradict results from the same region where the authors [57] found significantly higher $Na^+$ concentrations in degraded pasture areas, compared to areas with native babassu palm forest.

The strong correlation of CLFI-II with pH values and the concentrations of SOM and $Na^+$ resulted in the formation of a distinct group between this management system and other land-use systems (CLFI-I, intercropping, and pasture). Likewise, the strong association of pasture with high values of N, $K^+$, $Ca^{2+}$, SB, and CEC made this management system stand out from the others. Both pasture and CLFI-II had in common the presence of plant species established for several years (pasture with 17 years of adoption and adult babassu plants in CLFI-II). This is a relevant finding, considering that the sustainability of conservation land use systems has been associated with their adoption over long periods of time, generally longer than a decade [58–60]. This occurs because some significant changes in soil attributes can only be noticed after 10–20 years of its adoption [61,62].

Therefore, the data from this study suggest that for the eastern portion of the Brazilian Amazon, the establishment and maintenance of sustainable land use systems for longer periods of time constitute a determining factor for the consolidated improvement of soil fertility indicators and SOM accumulation. The data also demonstrate that the adoption of integrated systems, even the most complex ones containing eucalyptus as a tree component, did not result in short-term substantial improvements in the soil's chemical quality, which is the opposite of the initial hypothesis of this study. On the other hand, the adoption of systems that integrate the cultivation of grains with forage grasses in areas of natural occurrence of adult babassu palm plants proved to be a viable alternative, confirming the hypothesis that this native occurring species can be used as a tree component to replace eucalyptus. These results do not exclude the possibility of adopting recognized sustainable systems such as the CLFI-I system and the maize + Massai intercropping. However, such systems should be considered in long-term management plans in the region because they are known to have the potential to promote positive changes in soil attributes in tropical conditions similar to those of the present study [49,63]. To attest to the effectiveness of CLFI-I and maize + Massai intercropping for the conditions of the eastern Brazilian Amazon, further studies over longer periods of time should be considered. Future studies should also include the effects of the management systems on soil microbial attributes such as carbon and nitrogen microbial biomass, soil basal respiration, and enzyme activity, as these are sensitive indicators of soil quality changes.

## 5. Conclusions

The babassu palm can be used as a tree component in integrated systems, as it promotes increases in SOM concentration and soil pH and reduces attributes associated with acidity (H+Al and $Al^{3+}$) in the eastern region of the Brazilian Amazon. Such changes are associated with the babassu palm's ability to modulate the surrounding environment, giving the species a competitive advantage in anthropic environments and qualifying it as

a potential species for use in integrated systems in the region. In turn, pasture established over 17 years increases the values of N, $K^+$, $Ca^{2+}$, SB, and CEC. The data from this study suggest that the time of adoption is a determining factor in improving soil fertility attributes in the eastern portion of the Brazilian Amazon, and therefore, the consolidation of sustainable production systems in the region, including the tree component, must comply with long-term management plans.

**Author Contributions:** Conceptualization, J.B.C., H.A.d.S. and E.S.; methodology, I.M.d.S., J.B.C., R.C.F.F.P. and H.A.d.S.; software, I.M.d.S., D.C.d.S., H.A.F.d.A. and H.A.d.S.; validation, L.C.M., J.B.C. and H.A.d.S.; formal analysis, I.M.d.S. and H.A.d.S.; investigation, E.S., D.C.d.S., H.A.F.d.A., E.D.d.O.N., L.F.C.L., F.F.B., P.S.d.C.L. and H.A.d.S.; resources, J.B.C., R.C.F.F.P. and H.A.d.S.; data curation, E.S., J.B.C. and H.A.d.S.; writing—original draft preparation, I.M.d.S. and E.S.; writing—review and editing, I.M.d.S., E.S., J.O.L.d.O.J., F.F.B., P.S.d.C.L. and H.A.d.S.; visualization, M.D.M.A., L.C.M., J.B.C., D.C.d.S., H.A.F.d.A., E.D.d.O.N. and H.A.d.S.; supervision, J.B.C. and H.A.d.S.; project administration, J.B.C. and H.A.d.S.; funding acquisition, J.B.C. and H.A.d.S. All authors have read and agreed to the published version of the manuscript.

**Funding:** This research was funded by the Conselho Nacional de Desenvolvimento Científico e Tecnológico—CNPq (Grant number 311039/2017-0).

**Data Availability Statement:** Data will be made available upon request.

**Acknowledgments:** The authors are thankful to Banco da Amazônia—BASA, Empresa Brasileira de Pesquisa Agropecuária—EMBRAPA, Rede ILPF, Fundação Agrisus and Muniz Farm.

**Conflicts of Interest:** The authors declare no conflicts of interest.

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
