# Peer review of "Soil Chemical Quality in Integrated Production Systems with the Presence of Native and Exotic Tree Components in the Brazilian Eastern Amazon"

_forests, doi:10.3390/f15071078_

Round 1

Reviewer 1 Report

Comments and Suggestions for Authors

The authors surveyed the soil of 4 types of agricultural systems in East Amazon area. The authors found that the system ICLF-2, with tree component babssu palms, improved the soil property the most. The authors concluded that babassu palms can be used as tree component in integrated systems.

Major comments:

1. The authors described the experimental period as January 2017 to Feburary 2018. However, I found that in the authors said that the tree component in ICLF-I was introduced in Feb. 2016. Is this a typo? Different treatment time may change the outcome. I also wonder if pre-treatment soil samples were available for comparison.

2. The authors described the overall location of the experiment, but the location and layout of different treatments are not described. Were the treatments geographically separated?

Minor comments:

Line 50: the superscript of the unit is not properly applied.

Line 397: the font size changed unexpectedly.

Author Response

Dear reviewer,

Thank you for your important contributions. We made an effort to include in the manuscript all the suggestions, as specified below:

The authors surveyed the soil of 4 types of agricultural systems in East Amazon area. The authors found that the system ICLF-2, with tree component babssu palms, improved the soil property the most. The authors concluded that babassu palms can be used as tree component in integrated systems.

Major comments:

The authors described the experimental period as January 2017 to Feburary 2018. However, I found that in the authors said that the tree component in ICLF-I was introduced in Feb. 2016. Is this a typo? Different treatment time may change the outcome.

The correct year is 2017.We corrected the information.

I also wonder if pre-treatment soil samples were available for comparison.

We agree that this would be an important information. Unfortunately, there were no pre-treatment samples and original soil samples are no longer available

  1. The authors described the overall location of the experiment, but the location and layout of different treatments are not described. Were the treatments geographically separated?

We provided further information detailing size of areas and informing that areas were adjacent in the field.

Minor comments:

Line 50: the superscript of the unit is not properly applied.

We adjusted the information

Line 397: the font size changed unexpectedly.

We adjusted the font size

Reviewer 2 Report

Comments and Suggestions for Authors

In lines: 138, 298, 299, 302, 339, 345, 365, 366, should be replaced the name of the authors from references with an impersonal address or with expressions such as: the researches made by...., or other researches releaved that.....

Author Response

Dear reviewer,

Thank you for your important contributions. We made an effort to include all suggestions in the manuscript, as specified below:

- In lines: 138, 298, 299, 302, 339, 345, 365, 366, should be replaced the name of the authors from references with an impersonal address or with expressions such as: the researches made by...., or other researches releaved that.....

We replaced the names of authors by impersonal terms, as suggested

Reviewer 3 Report

Comments and Suggestions for Authors

I reviewed the article "Soil chemical quality in integrated production systems with the presence of native and exotic tree components in the Brazilian Eastern Amazon" by Souza Marques and co-authors.

The article is well-written and understandable.

I listed here some flaws of this manuscript:

A table in the text or as supplementary material with all the data is missing.

Line 70. The Authors mentioned the "integrated production systems", however, a proper description of such methods is missing or they are not clear enough

Line 133. Specify the reason for the dolomitic limestone application.

Line 147. Specify the amount (i.e, weight) of composite soil sample

Figure 4D: I think the x-label should report "Na+"

Figure 7: I prefer that also samples are plotted in PCA plots.

Lines 279-281: this sentence should be rephrased, as it is more appropriate for the conclusion section. As you are at the beginning of the Discussion section, you should explain as you "demonstrated that the integrated crop-livestock-forest system with babassu palm as a tree component was more efficient in increasing soil pH". For example you can refer to a Table or a figure.

Lines 286-290: this sentence is more proper to the introduction section to explain the advantages of Babassu palm rather than to add it in the Discussion section.

Line 317-319: explain better how the babassu palm can modify the pH of the soil

Through the entire manuscript:

- for the ions write the number of charge in superscript

- avoid specifying the abbreviation (e.g., SOM; CEC...) after you already did it once. For example, if you said that SOM means "soil organic matter" in the introduction section, you can avoid to repeat it again in line 157.

Author Response

Dear reviewer,

Thank you for your important contributions. We made an effort to include all suggestions in the manuscript, as specified below:

A table in the text or as supplementary material with all the data is missing.

We decided to present majority of data as Figures, for better visualization. We consider that presenting them also as Tables would imply in repetition of information. Moreover, numerical data (including raw data) can be made available if requested. Therefore, we kindly ask the reviewer to reconsider this suggestion.

Line 70. The Authors mentioned the "integrated production systems", however, a proper description of such methods is missing or they are not clear enough

We included a sentence in the Introduction section in order to describe the integrated production systems in the context of our study.

Line 133. Specify the reason for the dolomitic limestone application.

We inserted a sentence explaining that dolomitic limestone was used due to low Mg concentration in the experimental area. We also made clear that it was used based on soil analysis and recommendations for the region.

Line 147. Specify the amount (i.e, weight) of composite soil sample

We included the requested information five lines bellow, for better clarity

Figure 4D: I think the x-label should report "Na+"

Yes, we corrected the information

Figure 7: I prefer that also samples are plotted in PCA plots.

We changed the PCA Figure in order to make all samples evident as suggested. As we used this approach, we had also to recalculate new eigenvalues and eigenvectors, that are shown in Table 2.

Lines 279-281: this sentence should be rephrased, as it is more appropriate for the conclusion section. As you are at the beginning of the Discussion section, you should explain as you "demonstrated that the integrated crop-livestock-forest system with babassu palm as a tree component was more efficient in increasing soil pH". For example you can refer to a Table or a figure.

We removed the sentence and made changes in the whole paragraph in order to make information more straightforward.

Lines 286-290: this sentence is more proper to the introduction section to explain the advantages of Babassu palm rather than to add it in the Discussion section.

We rephrased the sentence in order to make clear that babassu palm pursues biomass characteristics that can lead to greater soil carbon concentration. We also moved some information to the Introduction section, as suggested.

Line 317-319: explain better how the babassu palm can modify the pH of the soil

We provided an attempt to explain babassu palm effects on soil pH in Lines 317 to 336.

Through the entire manuscript:

- for the ions write the number of charge in superscript

We included the valencies in superscript across the manuscript. The only exception was in the PCA (Figure 7), because R output does not provide it.

- avoid specifying the abbreviation (e.g., SOM; CEC...) after you already did it once. For example, if you said that SOM means "soil organic matter" in the introduction section, you can avoid to repeat it again in line 157.

We corrected the specification of abbreviations across the whole manuscript, as suggested.

Round 2

Reviewer 3 Report

Comments and Suggestions for Authors

The authors fulfilled all my requests. No other issues are detected. Therefore the article can be accepted in the present form.